# Benefits from Incorporating Virtual Reality in Pulmonary Rehabilitation of COPD Patients: A Systematic Review and Meta-Analysis

Irini Patsaki *, Vasiliki Avgeri, Theodora Rigoulia, Theodoros Zekis, George A. Koumantakis and Eirini Grammatopoulou

Department of Physiotherapy, University of West Attica, 11521 Athens, Greece
* Correspondence: ipatsaki@uniwa.gr

**Highlights:**

**What are the main findings?**

- VR programs could be used to augment the therapeutic effect of PR in COPD patients.
- VR rehabilitation programs could be used for home based programs as they are safe and feasible.

**What is the implication of the main finding?**

- Different games and environments offer the ability to tailor the exercise regimen to patients' needs and ability providing a personalized rehabilitation.
- Gamification features could increase adherence and participation of COPD patients.

**Abstract:** Chronic Obstructive Pulmonary Disease (COPD) is characterized by irreversible airflow limitation. Patient participation in Pulmonary Rehabilitation (PR) programs has a beneficial effect on disease management, improving patients' functional capacity and quality of life. As an alternative to traditional programs or as a complementary activity, the inclusion of virtual reality (VR) games is proposed. The aim of this research study was to investigate the effectiveness of incorporating VR in the pulmonary rehabilitation program of patients with COPD. A systematic literature search was performed for randomized controlled trials (RCTs) in the electronic databases Google Scholar, PubMed and Pedro from January 2014 to March 2022. The search involved screening for studies examining the effectiveness of enhancing PR with VR. The PEDro (Physiotherapy Evidence Database) scale was chosen as the tool to assess the quality of studies. A meta-analysis was performed where possible. Six studies were included in this systematic review. The PEDro scale showed five studies of good methodological quality and one of fair quality. The variables examined were: aerobic capacity for exercise, lung function, anxiety and depression, with non-significant improvement for the MRC Dyspnea scale, marginally non-significant improvement regarding 6MWT ($p = 0.05$) and significant improvement for FEV1 ($p < 0.05$). There was variability noted in the VR applications and the proposed rehabilitation that the experimental groups followed. The application of VR is recommended in COPD patients, in combination with conventional PR. VR was found effective in increasing the therapeutic effect and should be considered as a mean of increasing accessibility to PR. Therefore, further research, as well as additional RCTs regarding the effectiveness of VR in patients with COPD, seem necessary.

**Keywords:** virtual reality; exergaming; COPD; pulmonary rehabilitation; exercise capacity



## 1. Introduction

Chronic obstructive pulmonary disease (COPD) is a progressive lung disease and has been recognized as the third leading cause of death globally [1]. It represents a significant and growing healthcare concern as a leading cause of death and disability worldwide [2].

Affecting various functional and structural domains of the lungs, it has serious effects on the wider aspect of patients' well-being. Even younger patients show significantly diminished quality of life, manifesting a strong relationship of COPD with frequent exacerbations, comorbidities, and poor physical activity [3]. A recent study has detected a decrease in the burden of COPD that could be explained by the multiple strategies that have been implemented in recent years, such as the restriction of tobacco use, raising public awareness about prevention, and self-management of the disease [4]. Pulmonary rehabilitation programs are an integral part of the management of COPD and integrated intervention. Physical exercise is also an essential element, as it has a significant positive effect on the aerobic capacity, dyspnea, and health-related quality of life (HRQL) [5]. One of the main barriers for COPD patients to attend and complete their treatment and pulmonary rehabilitation programs are accessibility and lack of adherence [6].

Home-based pulmonary rehabilitation is a promising alternative model to improve uptake and access, as highlighted during the COVID pandemic. It has been well-presented that home-based PR is safe and may improve clinical outcomes with minimal resources [7]. New digital technologies offer unique opportunities to implement PR programs at home that ae tailored to each patient's specific needs [8]. Virtual Reality (VR) and Active Video Games (AVG) have appeared as an innovative technological solution to provide exercise at home, which could actively contribute to improved access and possibly adherence to a more active lifestyle [9]. Moreover, it is well-documented that they could provide high training loads. By selecting different modalities, especially by means of wii-fit, we could create a well-suited training load even for the most fragile patients [10]. Colombo et al. [9] defined VR as "a three-dimensional computer representation of a reality, which may be similar or completely different from reality, in which the participant can move physically while receiving multisensory stimulation". VR has used the term "gamification" from online gaming platforms as an attempt to integrate the positive features of gaming into virtual environments, applying it to various fields such as education, marketing, and health [11]. It derives from video games, as it embodies many of their elements, but differs from them as it does not have an exclusive entertainment purpose [12]. It has been established that gamification is a method that helps the user to find motivation and practice it through interactive techniques [13,14]. The combination of exercise and fun is crucial for a COPD patient, as endurance exercise is a major part of their rehabilitation in a lifelong program, which aims to prevent the condition from worsening [15]. Additionally, VR has been found to have a positive effect in mood and emotional state [16]. COPD patients need to follow an active lifestyle and exercise regularly in order to maintain the benefits of traditional pulmonary rehabilitation [17]. While this is not often possible due to various reasons, technology could assist. Therefore, this systematic review and meta-analysis intended to explore the effectiveness of implementing Virtual Reality in the pulmonary rehabilitation of patients with COPD as a means to enhance the delivered exercise regimen.

## 2. Methods

### 2.1. Study Design

A systematic review and a meta-analysis were conducted according to the Preferred Reporting Items for Systematic Reviews and Meta-Analyses (PRISMA) 2020 guidelines [18], and the methodological quality assessment of the clinical trials was conducted according to the PEDro scale [19].

The search was performed in the following online databases—Google Scholar, PubMed, and PEDro—from January 2014 to March 2022. During the search, the following keywords regarding the virtually applied intervention—"Virtual Reality" OR "Augmented Reality" OR "Video Games" OR "exergames" OR "serious games"—in combination with terms related to the pathology that was examined—"COPD" OR "Chronic Obstructive Pulmonary Disease" AND "Pulmonary Rehabilitation"—were used to create the different search strategies.

*2.2. Eligibility Criteria*

The criteria for inclusion of studies in this systematic review and meta-analysis were as follows: (a) RCTs (b) participants of 18 years of age or older; (c) participants with COPD of any severity; (d) in all studies the intervention group must have followed a VR training component added to PR and (e) be written in English.

Exclusion criteria from the research study were: (a) The control group having VR as an intervention method; (b) systematic reviews, publications of session lectures, study protocols, posters, cohort studies, case studies and abstracts, as they cannot be studied systematically; (c) other pulmonary diseases; (d) studies where the intervention does not cover the range of Pulmonary Rehabilitation interventions.

*2.3. Data Extraction and Quality Assessment*

A thorough review of the titles and abstracts of studies published in the databases used was performed. For those studies that met the criteria according to title and abstract, a full analysis was performed for further content review. Additionally, the reference lists of pertinent literature were searched for potentially relevant articles in English. The search strategy was carried out by two independent authors (A.V. and R.T.) and any differences were resolved by consensus between the two reviewers or by a third (I.P) when needed.

A predesigned data extraction form was used to extract the following data from the included articles: author, year of publication, sample size, a brief presentation of the type of the intervention that was used in each article and group, outcomes, and the differences reported between the two groups and within each group. Another table included the characteristics of the interventions, which were the type of intervention, the description of each game system that was used, the program duration, the frequency, and the session duration.

The methodological quality of the included studies was independently assessed by both investigators and any differences were resolved by a consensus. The PEDro (Physiotherapy Evidence Database) scale, which is valid and reliable [19,20], was chosen as the tool for assessing the methodological quality of the studies in this systematic review. It contains 11 criteria, 10 of which are answered with a yes or no response. If the criterion is satisfied, it is scored as 1 point, if not it is scored as 0. Criterion 1 affects external validity and does not contribute to the final PEDro scale score. 'Low quality' studies were defined as those scoring zero to three points, while studies were defined as 'moderate quality' and 'high quality' if they scored four to six points and seven to ten points, respectively [20].

*2.4. Data Synthesis and Analysis*

The Review Manager software (RevMan v.5.4.1) was used to summarize the effects of VR-Training on exercise capacity (6 Minute Walk Test-6MWT), subjective feeling of dyspnea (Medical Research Council, MRC scale), and pulmonary function (FEV$_1$%). Subgroup analysis was performed for each outcome if there was clinical heterogeneity in the intervention and other details of studies, such as the population characteristics. Studies were not categorized based on the follow-up time points, since all included studies analyzed the short-term effectiveness of VR-Training, comparing pre-intervention period and post-intervention period between the group differences.

Quantitative synthesis was carried out in accordance with the Cochrane Handbook for Systematic Reviews of Interventions guidelines, using the pre-post means and standard deviations from each chosen study for the between-group comparisons, either extracted directly from the articles or calculated where necessary [21]. Since the studies employed the same outcomes for the reported comparisons, the mean difference (MD) and 95% confidence intervals (CI) were used. To determine the clinical relevance of the treatment for each outcome, a random-effects inverse variance model was chosen for meta-analysis. The I$^2$ statistic was used as a measure of heterogeneity, with values greater than 50% interpreted to indicate significant heterogeneity [22].

## 3. Results

### 3.1. Identification of Studies

We identified 798 records through electronic database research. After excluding non-RCTs through electronic filters (*n* = 324) and duplicates (*n* = 18), we screened titles and abstracts from the remaining records. Having excluded studies that did not involve COPD participants (*n* = 177), non-VR intervention (*n* = 105), and studies written in a language other than English (*n* = 155), we thoroughly screened the remaining 19 in terms of full text inspection. After excluding 13 studies for not involving PR in both experimental groups, a total of six RCTs were finally included in this systematic review. A detailed flowchart is provided in Figure 1.

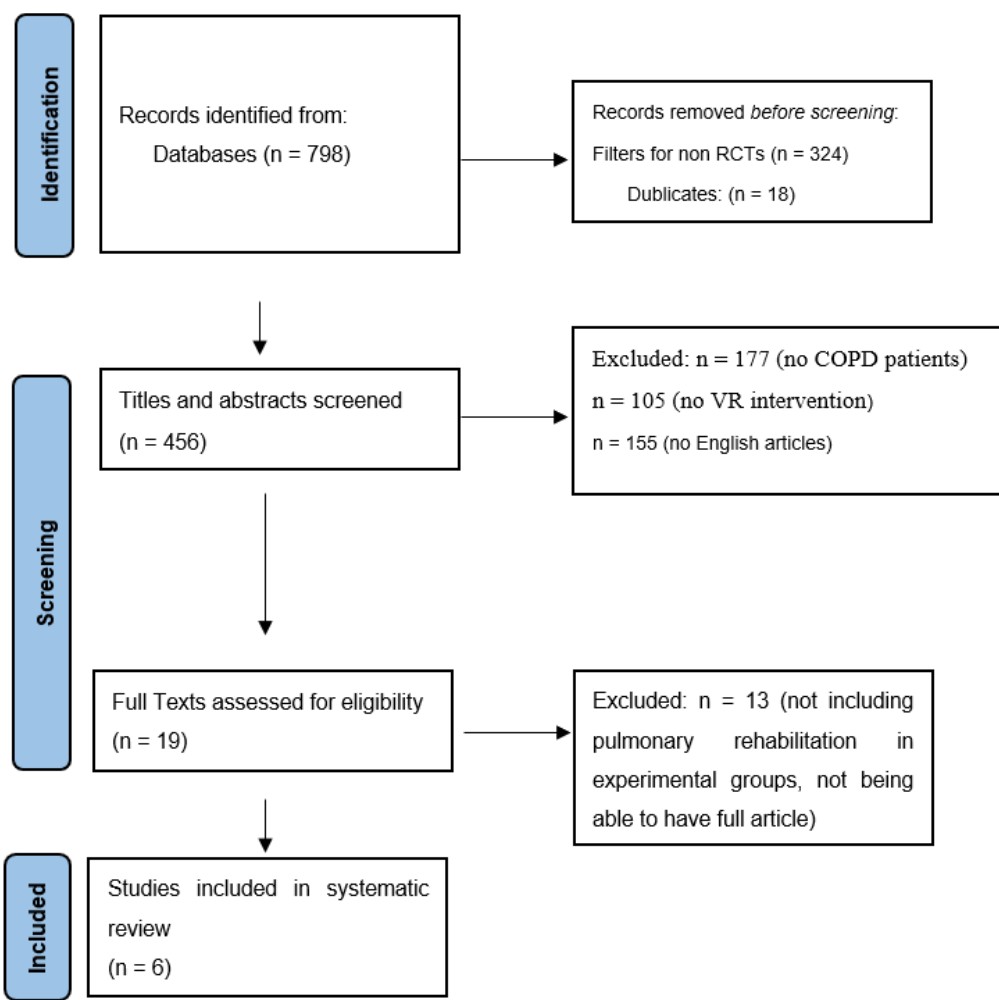

**Figure 1.** Prisma flow diagram.

### 3.2. Methodological Quality

The methodological quality score of all included studies was rated with the PEDro scale (Table 1) and, on average, was found to be 6.5/10. Specifically, two studies were rated with 8/10, two with 6/10, one with 7/10, and one with 4/10.

To address the risk of bias through the methodological quality of the included studies, we examined the 10 components of the PEDro scale individually (Figure 2). Only two categories—therapist and assessor blinding—were not addressed by all the studies. The measurement of outcomes obtained from >85% of subjects receiving treatment as allocated was not addressed by more than 50% of the included studies. These present significant sources of bias [23].

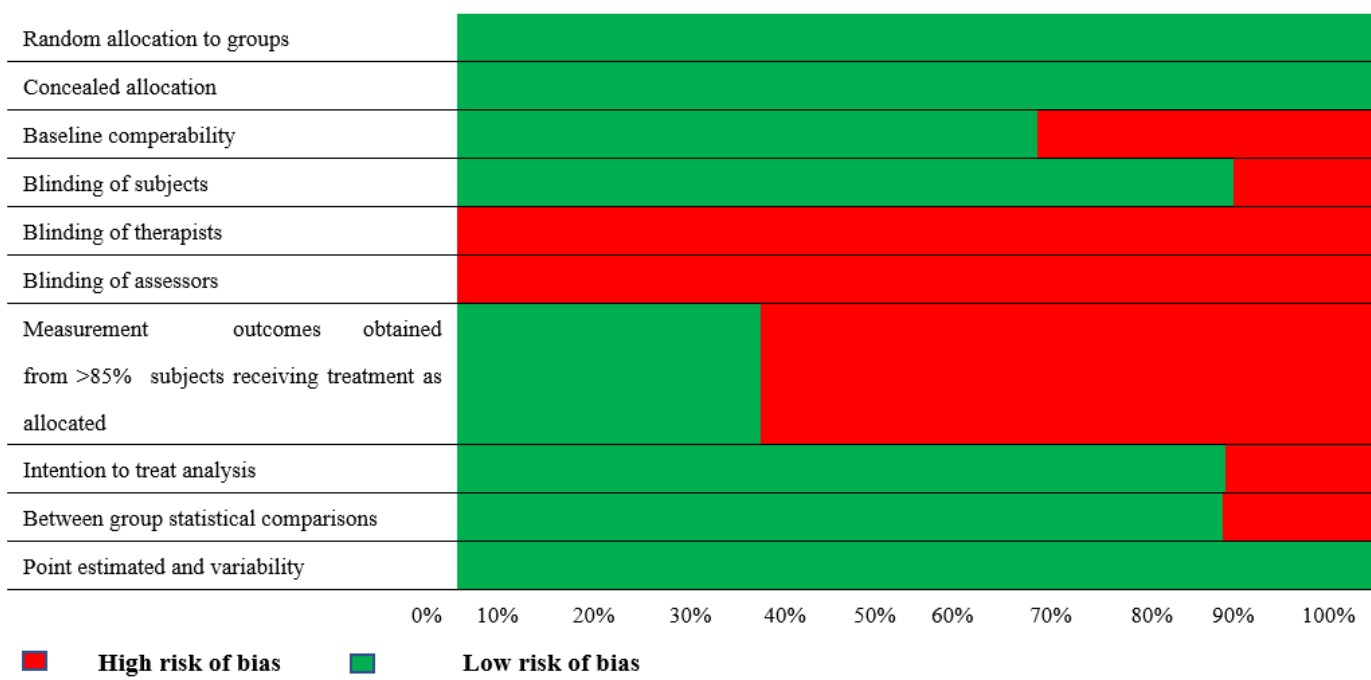

**Figure 2.** Resulting risk of bias per methodological quality item assessed with the PEDro scale.

**Table 1.** Rating of the included studies according to the PEDro scale.

| Criteria Studies | 1 | 2 | 3 | 4 | 5 | 6 | 7 | 8 | 9 | 10 | 11 | Score | Quality |
|---|---|---|---|---|---|---|---|---|---|---|---|---|---|
| Mazzoleni et al. (2014) [24] | 1 | 1 | 0 | 1 | 0 | 0 | 0 | 1 | 1 | 1 | 1 | 6/10 | Good |
| Sutanto et al. (2019) [25] | 1 | 1 | 0 | 1 | 0 | 0 | 0 | 0 | 0 | 1 | 1 | 4/10 | Fair |
| Xie et al. (2021) [26] | 1 | 1 | 1 | 0 | 0 | 0 | 0 | 1 | 1 | 1 | 1 | 6/10 | Good |
| Rutkowski et al. (2019) [27] | 1 | 1 | 1 | 1 | 0 | 0 | 0 | 1 | 1 | 1 | 1 | 7/10 | Good |
| Rutkowski et al. (2020) [28] | 1 | 1 | 1 | 1 | 0 | 0 | 1 | 1 | 1 | 1 | 1 | 8/10 | Good |
| Rutkowski et al. (2021) [15] | 1 | 1 | 1 | 1 | 0 | 0 | 1 | 1 | 1 | 1 | 1 | 8/10 | Good |

*3.3. Description of Studies*

The total number of participants in this systematic review was 360. All participants presented with stable COPD, apart from 16 in the study of Mazzoleni et al. [24], who presented with other pulmonary diseases. The mean age ranged from 64 to 75 years. In most studies FEV1% was >65, describing patients with a moderate degree of air obstruction. Only Suntanto et al. [25] and Xie et al. [26] included people with more severe COPD (FEV1% < 50, GOLD stage: D).

All included studies (Table 2) had added a VR component to usual PR, which was the main rehabilitation strategy for COPD patients. The technological equipment varied across the studies from the Microsoft Xbox Kinect [27,28] and Nintendo Wii Fit [24,25] to head-mounted displays [15,26].

**Table 2.** Characteristics of the studies included in the systematic review.

| Studies | Sample | Interventions | Control Group | Results |
|---|---|---|---|---|
| Mazzoleni et al. (2014) [24] | 39 CG: 19 EG: 20 | Wii Fit Plus System + PR | PR | 6MWT: EG vs. CG ($p = 0.028$) MRC dyspnea: EG vs. CG ($p = 0.488$) SGRQ: EG vs. CG ($p = 0.657$) BDEI: EG vs. CG ($p = 0.724$) STAI: EG vs. CG ($p = 0.788$) |
| Rutkowski et al. (2019) [27] | 68 CG: 34 EG: 34 | Kinect training + PR | PR | 6MWT: CG vs. EG ($p > 0.05$) |
| Rutkowski et al. (2020) [28] | 106 CG: 34 EG1: 38 EG2: 34 | EG1: Kinect training + PR + Stationary cycle ergometer EG2: Kinect training + PR | PR + Static cycle ergometer | 6MWT: EG1 vs. CG ($p = 0.011$) EG2 vs. CG ($p = 0.031$) |
| Rutkowski et al. (2021) [15] | 50 CG: 25 EG: 25 | Immersive VR + PR | PR + Schultz autogenic training | 6MWT: EG vs. CG ($d = -0.074$) $FEV_1$% pred: EG vs. CG ($d = -0.066$) HADS: EG vs. CG ($d = -1.175$) |
| Sutanto et al. (2019) [25] | 23 CG: 11 EG: 12 | Wii Fit System + Cycle Ergometer | Cycle Ergometer | 6MWT: EG vs. CG ($p = 0.226$) SGRQ: EG vs. CG ($p = 0.523$) MRC dyspnea: EG vs. CG ($p = 0.036$) |
| Xie et al. (2021) [26] | 60 CG: 30 EG: 30 | VR + PR | PR | Self efficacy score ($p < 0.05$) mMRC ($p > 0.05$) $FEV_1$%pred ($p > 0.05$) |

Acronyms: VR: Virtual Reality, PR: Pulmonary Rehabilitation, CG: Control Group, EG: Experimental Group, EG vs. CG: Statistically important difference using the *p* value or d as an indicator, 6MWT: 6 Minute Walk Test, BDEI: Beck Depression Inventory, BODE: body max index, airflow obstruction, dyspnea, and exercise capacity, $FEV_1$%pred: Forced Expiratory Volume in 1 s, GAD7: General Anxiety Disorder Scale, HADS: Hospital Anxiety Depression Scale, MRC: Medical Research Council, PSQ: Perceived Stress Questionnaire, SGRQ: Saint George's Respiratory, *p*: Statistically important difference is measured with *p* value, considered to be important as $p < 0.05$, d: The size of the between group effects was determined by Morris effect size d and classified as follows: 0.1–0.3; small effect; 0.3–0.5; intermediate effect and ≥0.5; strong effect.

The Xbox 360 console was used along with the Kinect motion sensor in order to detect and follow the participants' movements. The patients participated in mini-games as part of Kinect Adventures, such as rafting, cross country running, hitting a ball, and a roller coaster ride. The games that the Kinect training included were focused on improving balance, elasticity, endurance, and strengthening upper and lower limbs. Age-predicted maximal heart rate was used to monitor workload in order to ensure safe training. The Nintendo Wii Fit system uses haptic controllers and a balance board as interfaces to the games. In the study of Mazozoleni et al. [24], these involved a "Yoga" activity with a deep breathing session in a standing position on the balance board, the "Jogging Plus" that involved running on a spot, and "Twisting and squatting" that consisted of trunk twisting and arm–leg squatting. Similar games were used by Suntanto et al. [25], such as "Yoga", "Torso twist", and "Free run". Pulse rate, respiratory rate, and SpO2 were used to keep training safe, whilst intensity was monitored by using the 10-point Borg scale. Patients were instructed to maintain the sensation of dyspnea between 4–6 on the modified Borg scale.

In most recent studies, head-mounted displays (HMDs) have been used in order to immerse patients in a virtual environment. Thus, from semi-immersive gaming platforms we reached to explore the effectiveness of fully immersive gaming, either in the form of a simulated bicycle [26] or as a therapeutic garden that represents patient's health [15]. There are endless possibilities for this technologically revolutionary equipment. A full description of the rehabilitation programs, followed by the experimental groups, are individually presented for each group (Table 3).

**Table 3.** Description of the rehabilitation program of the experimental group.

| Study | Intervention | Program Duration | Frequency | Session Duration |
|---|---|---|---|---|
| Mazzoleni et al. (2014) [24] | (1) PR:<br>  (a) Exercise on a treadmill, a cycle, and an arm ergometer<br>  (b) Abdominal, upper- and lower-limb muscle activities involving lifting of progressively increasing light weight and shoulder and full-arm circling<br>  (c) Education<br>  (d) Nutritional programs and psychosocial counseling<br>(2) Wii Fit Plus:<br>  (a) "Yoga": Two 5-min sessions at the beginning and at the end of session, deep breathing in standing position on the balance board while maintaining the body's balance<br>  (b) "Jogging Plus": 10 min running on the spot<br>  (c) "Twisting and squat": 10 min trunk twisting and arm and leg squatting | 2 weeks PR + 1 week PR and Wii fit plus | Daily | (1) 30 min<br>(2) 1 h |
| Sutanto et al. (2019) [25] | (1) Exercise training on a cycle ergometer<br>(2) Wii Fit:<br>  (a) Yoga with "deep breathing" and "half moon", breathing techniques, holding particular poses for 10 min<br>  (b) "Torso twist": strength training<br>  (c) "Free run": running on spot while keeping the connected Wii Remote, which acted as a pseudo-pedometer | 6 weeks | 3 times per week | (1) 30 min<br>(2) 30 min |
| Rutkowski et al. (2019) [27] | (1) PR:<br>Physical capacity training, breathing exercises, inspiratory muscle training, inhalations, relaxation<br>(2) "Kinect training":<br>4 "minigames" ofτου "Kinect Adventures": "20,000 Leaks, Curvy Creek, Rally Ball, Reflex Ridge". | 2 weeks | 1 time per week | -- |
| Rutkowski et al. (2020) [28] | (1) PR:<br>  (a) Fitness exercises, coordination and balance, stretching exercises using elastic tapes, rehabilitation balls, and sensory cushions<br>  (b) Specific respiratory exercises for 30 min (relaxation exercises for breathing muscles, strengthening exercises of the diaphragm with resistance, exercises to increase costal or chest breathing, prolonged exhalation exercise, and chest percussion)<br>  (c) Group walks with a physiotherapist around the hospital<br>  (d) Inhalation with a NaCl isotonic solution<br>  (e) Rapidly changing postural drainage positions and chest percussions by a physiotherapist<br>  (f) Relaxation training, 15 min a day<br>(2) "Kinect training":<br>same minigames as Rutkowski et al. (2019)<br>(3) Endurance exercise training with a stationary cycle ergometer | 2 weeks | 5 times per week | (1) 15–30 min each exercise<br>(2) 20 min<br>(3) 20–30 min |
| Rutkowski et al. (2021) [15] | (1) PR: Same as Rutkowski et al. (2020)<br>(2) VR with "HMD": The software features a Virtual Therapeutic Garden which is a metaphor for the patient's health: at the beginning it appears as untidy and grey, yet with each session it becomes more alive, symbolizing the process of recovery | 2 weeks | 5 times per week | (1) 15–30 min each exercise<br>(2) 20 min |
| Xie et al. (2021) [26] | (1) PR:<br>Participating in a disease explanation activity, breathing exercises<br>(2) VR:<br>"HMDs", data gloves, simulated bicycle | 8 weeks | -- | (1) 35 min<br>(2) 20 min |

Acronyms: VR; Virtual Reality, PR; Pulmonary Rehabilitation, HMD: Head-Mounted Display.

*3.4. Intervention Comparability*

All of the included studies were randomized, included a control group, and had an adequate number of individuals. Only one study had a relatively low number of participants [25], with most ranging between 20 to 30 per group. A sample size calculation was performed in all studies.

Although significant clinical heterogeneity was noted between the included studies attributed to: (a) differences in the technology used and variability in (b) intervention duration, (c) type of exercise, and (d) the outcomes assessed between studies, a quantitative synthesis was also performed where possible.

*3.5. Effects of Interventions*

3.5.1. Effect of VR Training on Exercise Capacity (Figure 3)

The effect of VR-Training with or without other parallel interventions on 6MWT, calculated in meters, was evaluated in four studies [24,25,27,28], including 196 participants in total (Figure 3). A mean difference (MD) (95% CI) = 15.93 (−0.14 to 31.99) m, favoring VR-Training with marginal non-statistical significance (Z = 1.94, *p* = 0.05) and substantial heterogeneity ($I^2$ = 72%, *p* = 0.01) was noted, based on an 8/10 Pedro quality score on average (Table 1).

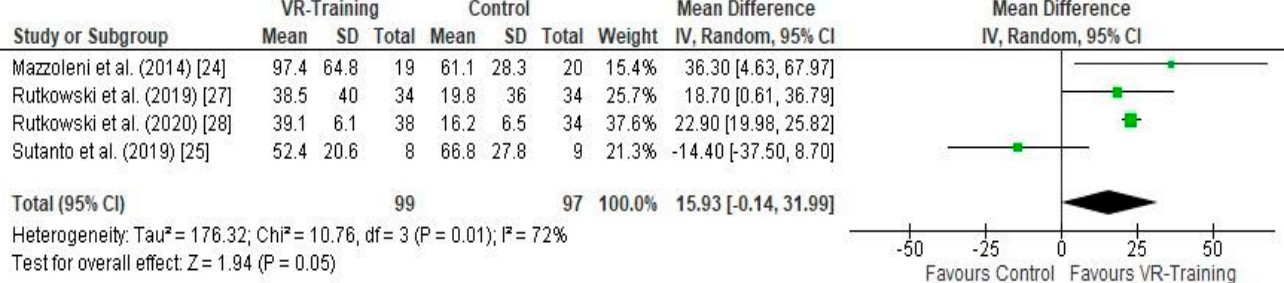

**Figure 3.** Forest plot showing the effects of VR-Training on the 6MWT [24,25,27,28].

3.5.2. Effect of VR Training on Pulmonary Function (Figure 4)

The effect of VR training with or without other parallel interventions on $FEV_1$%pred was evaluated by two studies [15,26], including 110 participants in total (Figure 4). A mean difference (MD) (95% CI) = 4.56% (1.64 to 7.49), favoring VR training with statistical significance (Z = 3.06, *p* = 0.002) and minimal statistical heterogeneity ($I^2$ = 4%, *p* = 0.31) was noted, based on Pedro-quality evidence (Table 1).

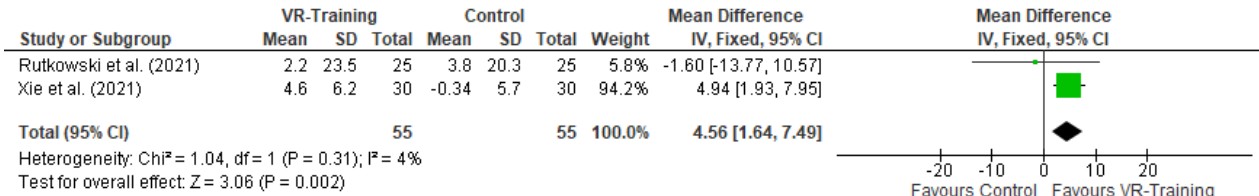

**Figure 4.** Forest plot showing the effects of VR training on the $FEV_1$%pred [15,26].

3.5.3. Effect of VR Training on Subjective Dyspnea (Figure 5)

The effect of VR-Training with or without other parallel interventions on the MRC dyspnea scale was evaluated by two studies [24,25], including 56 participants in total (Figure 5). A mean difference (MD) (95% CI) of −0.15 (−0.45 to 0.15) was found, with both studies favoring VR-Training, but overall not reporting statistical significance (Z = 1.00, *p* = 0.31); however, no statistical heterogeneity ($I^2$ = 0%, *p* = 0.74) was noted based on 5/10 Pedro quality evidence on average (Table 1).

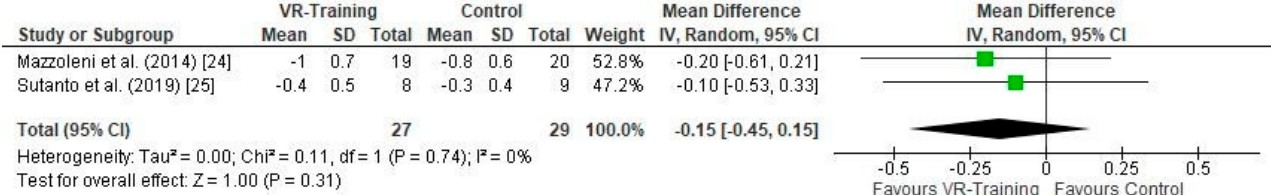

**Figure 5.** Forest plot showing the effects of VR-Training on the MRC dyspnea scale [24,25].

### 3.5.4. Psychological Status

Anxiety and depression were evaluated in two studies [15,24] without reaching statistically significant changes between groups, although a significant reduction was noted in the VR intervention in both studies.

## 4. Discussion

The aim of this systematic review and of the accomplished meta-analysis was to provide novel information from well-designed RCTs regarding the benefits that emerge from adding VR technology to traditional PR of COPD patients. Technological innovations push the limits of rehabilitations to new boundaries. Incorporating gamification features into traditional rehabilitation programs through VR gaming or exergaming have been presented with promising results in different populations [29,30].

One of the main findings from this review was the marginal positive effect that these novel technologies have on exercise capacity. The heterogeneity noted among the training modalities could have limited the overall effect. Yet, this is a key outcome that is measured in all respiratory patients after completing a PR program. There were three studies that showed significant difference among experimental groups in 6MWT [24,27,28], with all managing to exceed the minimal clinical importance difference of 35 m. All three included games that were focused on improving dynamic balance, strengthening the lower and upper limbs and improving endurance, thus providing an extra stimulus. Only [28] Rutkowski et al. did explore and showed that even the use of Kinect-based training alone could lead to a significant improvement of 6MWT. It could be assumed that AVG could induce high physiological demands capable of producing significant training effects if used regularly as a training method [31] (Kuys et al., 2011). In the studies included in this systematic review, the intensity of training was monitored using the Borg dyspnea scale and ranged in similar values (4–6 points), as recommended by the ATS (American Thoracic Society) guidelines for pulmonary rehabilitation for COPD patients [32]. LeGear et al. [33] managed to show that the level of physical effort produced during VR rehabilitation with NintedoWii was similar to that produced during training on a treadmill. With the exception of one study [15], all the others tried to incorporate either a strengthening or an endurance training component or even both through the gaming intervention, following the ATS guidelines regarding COPD rehabilitation. Even the ability to provide a training session safely is of significant importance. We should bear in mind that a well-designed program tailored to the abilities and needs of the patients even when games are utilized as part of the training regimen could have positive results. Being able to improve exercise capacity and thus increasing physical activity is a key component to reducing exercise intolerance and providing a healthier and more active lifestyle for this population. Also, the advantages that gaming brings in motivation, engagement and pleasure are important components of any exercise-based intervention. Additionally, we should add the ability of delivering this service remotely, thus overcoming a lot of barriers that patients mention when dropping out of PR [34]. Similar results were presented by a recent meta-analysis in which non-English studies were included [35].

Although beneficial effects were noted in spirometry values, dyspnea did not show a significant improvement. The results were inconsistent between the two studies that used MRC to assess dyspnea [24,25]. We should take into consideration the multifactorial cause of

dyspnea in COPD patients and its relationship with peripheral muscle dysfunction [32,36]. In the study of Mazzoleni et al. [25], the small duration of the VR intervention (1 week) did not present improvements in dyspnea or in functional ability, despite the positive effect in 6MWT. It is likely that interventions with a longer duration would be able to positively affect dyspnea. It is argued that the ability of these systems to achieve a level of physical exertion is similar to that of a traditional pulmonary rehabilitation program that is well-tailored to the patients' abilities [35].

Although it was not investigated by the majority of the studies, psychological status and the burden of both anxiety and depression is well-highlighted in COPD populations. Previous studies have also pointed out the possible positive effect that a more entertaining exercise program could have [37]. Reducing stress levels and mood swings may be a key feature to maintain interest and increase engagement in PR thus increasing the chance of long-term improvements. It is well-described that benefits that were obtained from PR were easily lost once physical activity was reduced and regular exercise was neglected [38]. Different ways of increasing adherence, such as telephone calls, activity monitors, or even cell phone applications, have been used in order to maintain the positive effects of PR [39,40]. However, the use of gaming platforms could provide us with accurate and precise information regarding the patients' adherence.

Introducing gaming features to rehabilitation and thus bringing pleasure, joy, socialization, and a competitive spirit could be the components that traditional PR needs in order to fully change the sedentary behavior of COPD patients. Several studies have shown that patients are interested in VR and present increased satisfaction [24,33,41–43]. VR interventions could be an option for ongoing self-exercise once a pulmonary rehabilitation program has been completed. Albores et al. demonstrated the positive effect of a home-based program in exercise capacity and quality of life and suggest the use of intervention in COPD patients that are unable to attend traditional programs [44].

Limitations of the study. A few limitations should be noted regarding the language restrictions and the variety of technological applications being included, along with the variability in outcomes measured. The difference in visualization modality between screens and HMDs affects the sense of presence and thus the interaction and physical performance. Physical engagement and the level of physical activity that could be produced certainly may differ between the different technological equipment.

## 5. Future Studies

Although active or exergames are increasingly being used in respiratory diseases, future studies need to explore and measure the physical exertion achieved in different tasks via VR training. Also, it would be most interesting to have follow-up data exploring adherence and behavior change over time, which are both important in these patients. We need to further explore our ability to offer unsupervised rehabilitation programs that have the same positive effects as supervised ones. This will allow us to reach out to populations that face serious problems in participating in traditional PR due to lack of time or increased distance. It would be most interesting to incorporate not only training regimens, but educational regimes that could be targeted to stress and anxiety management, as seen in Rutkofski et al. [15].

## 6. Conclusions

This systematic review and meta-analysis demonstrated that a VR program could be used to augment the therapeutic effect of PR in COPD patients, as it seems to have a beneficial effect in exercise capacity and on lung function. It is a safe and well tolerated intervention that offers adequate work loading to traditional PR, while being delivered at home and in one's spare time. Gamification features add enjoyment and create a spirit of competition that fuels the ongoing engagement. Further studies are needed to evaluate the effect of VR programs on other significant variables related to COPD wellbeing such as depression, anxiety, cognition and quality of life. Assessing the long-term utilization

of these programs and maintenance of positive results will be of high importance for this population and those with other respiratory conditions.

**Author Contributions:** Conceptualization, E.G. and I.P.; Methodology, I.P., G.A.K., V.A. and T.R.; Writing-original draft preparation, I.P., T.Z. and G.A.K.; Supervision, E.G. and I.P.; editing and reviewing, E.G. and G.A.K. All authors have read and agreed to the published version of the manuscript.

**Funding:** This research received no external funding.

**Institutional Review Board Statement:** Not applicable.

**Informed Consent Statement:** Not applicable.

**Data Availability Statement:** No new data was created.

**Conflicts of Interest:** The authors declare no conflict of interest.

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
