# Peer review of "Benefits from Incorporating Virtual Reality in Pulmonary Rehabilitation of COPD Patients: A Systematic Review and Meta-Analysis"

_arm, doi:10.3390/arm91040026_

Round 1

Reviewer 1 Report

This well-written paper is a systematic review and meta-analysis intended to explore the effectiveness of implementing VR in pulmonary rehabilitation of patients with COPD, as a means to enhance the delivered exercise regimen.

This reviewer is interested in the search terms used, for instance, did the authors also try "simulation", "serious games", or "active video games", or other terms? It seems there may have been studies missed.

This reviewer felt the study to be methodologically very sound. The main issue might be that only six studies ended up being fully analyzed. Their representativeness is possibly in question, even if they did meet all inclusion criteria.

A related concern--that is addressed by the authors--is the activity required by participants across the studies within VR. Given that equipment differed (a Wii is not really VR), details of what was required of participants also differed, although at a higher perspective they all involved exercise of some kind.

Overall, though, the authors do a nice job presenting an approach to analyzing an important feature of VR-based applications for a real-world need, and also do a nice job tying together their findings.

Author Response

We would like to thank the reviewer for his/her cumulative comments and suggestions, that could further improve our work.

-Indeed, the last few years with the development of new rehabilitation strategies exploring the use of gamification in health there are several terms that have been used as the reviewer suggested. We  avoided using the term simulation that targets other areas not related to pulmonary rehabilitation and we used the term “video games” and not “ active video games” in  order to allow more studies to be presented from the search. Serious games is a term widely used in neurorehabilitation and in terms of education. Yet, we used the suggested term as proposed by the reviewer in our search strategy, but without retrieving any study that met our inclusion criteria in order to be added. Thus, we included  the term in section 2.2 and included the number of studies (n=13) that were found in section 3.1 and made appropriate changes in figure 1.

 -We would like to underline that our inclusion criteria and our aim to investigate possible benefits from incorporating virtual reality rehabilitation strategies to traditional pulmonary rehabilitation is the cause for excluding studies that the reviewer may thought as missed or created the filling of limited representation. We tried to seek and present new knowledge on the field of pulmonary rehabilitation as it is the golden standard for this population. Thus, strategies that could increase effectiveness and probably increase participation and adherence are of significant value.

-This heterogeneity in the intervention being described under the umbrella term of VR is not something new in the field of rehabilitation and especially when technology is involved. We mentioned this and we will further highlight it in section 3.4 regarding intervention comparability by included another aspect of variability that of exercise type. Additionally, we added to the limitations at the end of the discussion “Physical engagement and the level of physical activity that could be produced certainly may differ between the different technological equipment” in order to underline this. Once, more studies are published on the issue we could be able to have a more clear picture by making comparisons between different equipment.

Reviewer 2 Report

  1. I wonder if the authors can state more clear the hypothesis which generated the aim of the research
  2. I wonder if the authors can enlarge their search also in the grey literature and in the references of the studied articles
  3. A benefit can be to present the concepts behind Virtual Reality and Augmented Reality connected  with Wii Fit exercise games (a start point can be from the articles: Celesti A, Cimino V, Naro A, Portaro S, Fazio M, Villari M, Calabró RS. Recent Considerations on Gaming Console Based Training for Multiple Sclerosis Rehabilitation. Med Sci (Basel). 2022 Feb 11;10(1):13. and Găină MA, Szalontay AS, Ștefănescu G, Bălan GG, Ghiciuc CM, Boloș A, Găină AM, Ștefănescu C. State-of-the-Art Review on Immersive Virtual Reality Interventions for Colonoscopy-Induced Anxiety and Pain. J Clin Med. 2022 Mar 17;11(6):1670. )
  4. The references are appropiate
  5. Conclusions might add the advantage of VR in COPD patients, compared with classical methods.  

Author Response

We would like to thank the reviewer for his/her recommendations and comments.

  1. Trying to clarify the hypothesis behind our aim for this study we added the following in the introduction  “COPD patients need to follow an active lifestyle and exercise regularly in order to maintain benefits of traditional pulmonary rehabilitation (Silva et al. 2022). Although, this is not often possible due to various of reasons, technology could assist. and the relevant reference “Silva, L., Maricoto, T., Costa, P., Berger-Estilita, J., & Padilha, J. M. (2022). A meta-analysis on the structure of pulmonary rehabilitation maintenance programmes on COPD patients' functional capacity. NPJ primary care respiratory medicine32(1), 38.”
  2. Our purpose wasn’t to search the grey literature as we wanted to included well designed randomized control trials and not to include other type of studies. Yet, we searched the references of all the studied included as mentioned at section 2.3 Data extraction and Quality assessment. We highlighted the specific sentence for the reviewer.
  3. We would like to thank the reviewer for suggesting further articles, yet there are significant differences between the features of neurorehabilitation and pulmonary rehabilitation and even psychological ones. Thus, we included literature from another respiratory disease, that is quite challenging in order to underline the fact that Wii fit exercise games are of high value for a rehabilitation program that need both strength and endurance elements as these are included in traditional pulmonary rehabilitation programs. This is presented in discussion, but we further included the following in introduction: “Moreover, it is well documented that could provide high training loads. By selecting different modalities, especially in means of wii-fit, we could create the most suited training load even for the most fragile patients (del Corral et al. 2014)” and in the discussion we added the following “ It is well presented that AVG could induce high physiological demands capable of producing significant training effects if used regularly as a training method (Kuys et al. 2011).” And “   LeGear et al. [33] managed to show that the level of physical effort produced during VR rehabilitation with Nintedo Wii, was similar to that produced during a training on a treadmill.”
  4. We included further studies as proposed by reviewers.
  5. We further enrich our results as kindly suggested by the reviewer. We added the following: “. It is a safe and well tolerated intervention that offers adequate work loading to traditional PR, while being delivered at home and at ones spare time. Gamification features adds enjoyment and creates a spirit of competition that fuels the ongoing engagement.”

Reviewer 3 Report

Dear authors, 

It is a very interesting work but the theoretical framework must be expanded with more references.

Likewise, they must adapt the article to the standards of the journals in terms of presentation of tables, citations, references, etc.

Regarding the search in databases, the systematic review is poor since only three databases are used. I suggest the authors broaden this search with databases such as Scopus, Web of Science, APA Psycinfo because it is likely that there are interesting articles that have not been included in this systematic review.

I also recommend calculating the risk of bias of the selected articles.

It is important to extend  the conclusions.

Author Response

We would like to thank the reviewer for his comments and suggestions. More references have been included (there are highlighted) as result of comments and suggestions that were made from other reviewers.

-When we submitted our manuscript, we did so without any adaptation to the journal’s format as this was an option. The manuscript has been revised in order to follow the instructions of the journal regarding citations, references, abstract presentation .

-Respectifully, we believe that a minimum of three databases is an acceptable number to be used especially, when between them there are two major ones. According to the PRISMA statement 2009 full electronic search strategy for at least one database should be presented, while according to the checklist of AMSTAR at least 2 databases should be searched along with at least one supplementary search.

-We used PEDrro scale to assess the methodological quality of the included RCT’s and the same tool for risk of bias, section 3.2 and figure 2. We believe that this is sufficient.

-Further enrichment of the conclusions was done by adding the following: “ . It is a safe and well tolerated intervention that offers adequate work loading to traditional PR, while being delivered at home and at ones spare time. Gamification features adds enjoyment and creates a spirit of competition that fuels the ongoing engagement.”

Round 2

Reviewer 3 Report

The paper can be accepted.